# Local maximum synchrosqueezes form scaling-basis chirplet transform

**Yating Hou**[1,2], **Liming Wang**[1,2]*, **Xiuli Luo**[1,2], **Xingcheng Han**[1,2]

**1** State Key Laboratory of Dynamic Testing Technology, North University of China, Taiyuan, China, **2** School of Information and Communication Engineering, North University of China, Taiyuan, China

* wlm@nuc.edu.cn

**Data Availability Statement:** The bat signal used in this paper is publicly available from the Rice University dataset (https://web.archive.org/web/20160403234536/ http://dsp.rice.edu/software/bat-echolocation-chirp). The vibration signals used

## Abstract

In recent years, time-frequency analysis (TFA) methods have received widespread attention and undergone rapid development. However, traditional TFA methods cannot achieve the desired effect when dealing with nonstationary signals. Therefore, this study proposes a new TFA method called the local maximum synchrosqueezing scaling-basis chirplet transform (LMSBCT), which is a further improvement of the scaling-basis chirplet transform (SBCT) with energy rearrangement in frequency and can be viewed as a good combination of SBCT and local maximum synchrosqueezing transform. A better concentration in terms of the time-frequency energy and a more accurate instantaneous frequency trajectory can be achieved using LMSBCT. The time-frequency distribution of strong frequency-modulated signals and multicomponent signals can be handled well, even for signals with close signal frequencies and low signal-to-noise ratios. Numerical simulations and real experiments were conducted to prove the superiority of the proposed method over traditional methods.

## 1. Introduction

In reality, many signals are nonstationary [1, 2], and the most common feature of these signals is that their frequency is a one-dimensional non-constant function with respect to a time variable. The frequency used in a Fourier transform, however, is an average description of the signal's frequency change over time. Nonstationary signals can no longer be analyzed using the conventional frequency approach [3–5]. To represent this crucial time-frequency characteristic of the change in the frequency of nonstationary signals with time, the concept of instantaneous frequency has been developed [6, 7]. An effective method for processing nonstationary signals is time-frequency analysis (TFA) [8–10]. The result is a time-frequency representation (TFR) in the form of a time-frequency-density function [11], which contains rich signal information, including the distribution pattern of signal energy in the time-frequency plane, instantaneous frequency characteristics, and instantaneous bandwidth. Using only conventional signal analysis techniques makes it challenging to perform accurate analyses because mechanical equipment have numerous excitation sources, complicated generation mechanisms [12], sound transmission paths, and has non-smooth, non-linear characteristics [13]. Consequently, fault detection ideas that were previously guided by signal analyses are now being extended [14].

in this paper are publicly available from the Case Western Reserve University dataset (https://engineering.case.edu/bearingdatacenter/apparatus-and-procedures).

**Funding:** The manuscript is funded by:(1)Science and Technology Innovation Project of Colleges and Universities in Shanxi Province(China),the award number is 2020L0301; (2) Fundamental Research Program of Shanxi Province(China), the award number is 20210302124545.

**Competing interests:** The authors have declared that no competing interests exist.

There are various TFA methods available at present. The concept of TFA methods originated from the Gabor expansion theory proposed by a Hungarian physicist, Gabor, in 1946. The famous linear time-frequency transform method, the short-time Fourier transform (STFT), was developed based on it. Classical TFA methods also include the Wigner-Ville distribution (WVD), wavelet transform (WT), and S-transform, which have allowed multi-resolution analyses of signals to be performed [15, 16]. Owing to the constraints of the Heisenberg-Gaber uncertainty principle, linear TFA methods should strike a balance between time and frequency resolutions. Bilinear time-frequency distribution (TFD) methods are also limited in applications dealing with multi-component analysis signals owing to the interference of the cross terms [17, 18]. The chirplet transform (CT) has been proposed to improve the energy concentration of the time-frequency map. The CT is a new TFA method that can be considered as a generalization of STFT and WT. However, when the frequency of a signal shows a nonlinear variation with time, the resolution of CT is low, and the accuracy of its analysis cannot be guaranteed. Therefore, polynomial chirplet transform (PCT) [19], modified spline-kernelled chirplet transform (MSCT) [20], velocity synchronous linear chirplet transform (VSLCT) [21], general linear chirplet transform (GLCT) [22], and scaling-basis chirplet transform (SBCT) [23] have been developed based on CT. However, the energy concentration of these TFA methods is not satisfactory, and they often exhibit poor noise resistance [9].

To overcome the aforementioned limitations, three researchers, Kodera, Gendrin, and Villedary, used the phase information of an analyzed signal to collect the scattered time-frequency energy in the time-frequency plane, which marked the development of the first time-frequency post-processing techniques. The reassignment method (RM) algorithm with a solid theoretical foundation was proposed as a post-processing TFA method. This method is mainly used to enhance the effect of time-frequency representation; however, it does not support signal reconstruction. Daubechies et al. then proposed the synchrosqueezing wavelet transform (SWT) in 2011, SWT rearranges the time-frequency coefficients using the synchrosqueezing operator, shifts the TFD of the signal at any point in the time-frequency plane to the center of gravity of the energy, and enhances the energy of the instantaneous frequency [24, 25]. This can solve the time-frequency ambiguity problem of traditional TFA methods. The synchroextracting transform (SET) and local maximum synchrosqueezing Transform (LMSST) were also proposed by Yu et al. [26, 27]. Unlike the classical synchrosqueezing transform theory, SET is only concerned with the instantaneous frequencies corresponding to the characteristic components of a signal. The divergent time-frequency energy coefficients are removed by the simultaneous extraction operator, and only the time-frequency ridge coefficients are retained. Tu proposed a horizontal synchrosqueezing transform (HST) that solves the problems of traditional SST by applying a local estimation of the group delay [28]. Zhu proposed a synchroextracting transform based on CT (SECT). SECT shows a better performance than certain advanced TFA methods [29].

With these applications of SST, we realize that it can be considered as a postprocessing technique based on the traditional TFA methods (STFT, WT, and CT). Fundamentally, the effectiveness of these postprocessing methods and the superiority of the processed signals also depend on the original TFA methods. The more accurate the instantaneous frequency obtained by the traditional TFA method, the clearer the separation of multi-component signals; therefore, postprocessing can play an important role in multicomponent signal analysis. However, once these traditional TFA methods have obtained the instantaneous frequency deviation coupled with the interference of noise, the post-processing results may become misleading. Therefore, it is necessary to first guarantee that the instantaneous frequency after TFA matches the actual instantaneous frequency of the signal. The SBCT method reconstructs a new chirplet, and this transform can match each instantaneous frequency slope in a

multicomponent signal within the same window length. Although higher energy concentrations can be achieved, the frequency resolution is not clear enough. To obtain a higher frequency resolution, this study extends the synchrosqueezing transform to SBCT and proposes a new TFA method called the local maximum synchrosqueezing scaling-basis chirplet transform (LMSBCT), which can suppress the interference of noise more effectively and has a higher time-frequency aggregation compared with the existing TFA methods, to obtain the TFR.

The remainder of this paper is organized as follows: In Section 2, the SBCT and LMSST methods are introduced. In Section 3, the theory of the LMSBCT method proposed in this study and its algorithm implementation are introduced in detail. In Sections 4 and 5, the superiority of the proposed algorithm proposed is discussed and demonstrated through simulation experiments and real cases. Finally, Section 6 concludes the study.

## 2. Theoretical principles

### 2.1 SBCT theory

The expression of CT is given by

$$CT(f, t_c) = \int_{-\infty}^{+\infty} s(u)h(u - t_c)\exp(-j2\pi\varphi(f, u, t_c))du \tag{1}$$

where, $s(u)$ denotes the Hilbert transform of the signal $x(u)$, $h(u)$ denotes the real even Gaussian window function, and $\varphi$ is a phase function defined as $\varphi(f, u, t_c) = fu + C(u - t_c)^2/2$. The second-order derivative of the phase function yields C: It follows that the rotation angle $\theta$ has a constant value.

$$\varphi'(f, u, t_c) = d\varphi/du = f + C(u - t_c) \tag{2}$$

$$d\varphi'/du = C = -\tan(\theta) \tag{3}$$

At one window length, when the instantaneous frequency trajectory of the signal changes with time, different C values are required to achieve a higher energy resolution, and when the signal has multiple components, different C values are also required to match the frequency trajectory of the signal simultaneously. To overcome the limitations of the traditional CT method for determining the chirp rate, SBCT reconstructs a new phase function.

$$\varphi_s(f, u, t_c, a_1, a_2, \ldots, a_n) = f \times (u + \sum_{k=1}^{n} a_k(u - t_c)^{1+k}) \tag{4}$$

The second-order derivative of this function gives the equation concerning $\theta$.

$$\varphi_s' = \frac{d\varphi_s}{du} = f \times (1 + \sum_{k=1}^{n} (1 + k)a_k(u - t_c)^k) \tag{5}$$

$$-\tan(\theta) = \frac{d\varphi_s'}{du} = f \times (\sum_{k=1}^{n} (1 + k)ka_k(u - t_c)^{k-1}) \tag{6}$$

When the signal is at the moment $u = t_c$, i.e. $u \in (t_c - \frac{L}{2}, t_c + \frac{L}{2})$, $\theta$ can be expressed as:

$$\theta = -\arctan(2fa_1) \tag{7}$$

that is, when the signal is a multicomponent signal. Simultaneously, different signal components have different $\theta$ values corresponding to the component time-frequency spine. Thus, the

time-frequency resolution will be greatly improved, and the energy concentration will be higher.

when the signal is at moment $t_c + \Delta u$, i.e. $\Delta u \in (-L/2, L/2)$,

$$\theta = -\arctan(f_c \times (\sum_{k=1}^{n} (1+k)ka_k(\Delta u)^{k-1})) \tag{8}$$

that is, when the signal is a strong time-varying signal, different moments will have various $\theta$ values to correspond with the time-frequency ridge.

Substituting the new phase function in Eq (4) into Eq (1), the SBCT can be expressed as

$$SBCT(f, t_c, a_1, a_2, \ldots, a_n)$$
$$= \int_{-\infty}^{+\infty} s(u)h(u - t_c)\exp(-j2\pi f \times (u + \sum_{k=1}^{n} a_k(u - t_c)^{1+k}))du \tag{9}$$

## 2.2 LMSST theory

SST is a TFA postprocessing technique that builds on the obtained TFD and uses the local behavior (phase information) near the time-frequency point to perform frequency rearrangement of the TFD. Its significant contribution is to increase the time-frequency aggregation and time-frequency ridges in more detailed.

For a signal to be measured, $x(u)$ should satisfy $f \in L^2(R)$. $|G(t,\omega)|$ denotes the spectrogram of the STFT, and SST is calculated as

$$T_{SST}(t, \eta) = \int_{-\infty}^{+\infty} G(t, \omega)\delta(\eta - \omega_0(t, \omega))d\omega \tag{10}$$

where, $\delta$ denotes the Dirac function and $\omega_0(t,\omega)$ can be obtained using the following equation:

$$\omega_0(t, \omega) = -i\frac{\partial_t G(t, \omega)}{G(t, \omega)} = \omega + i\frac{G^{h'}(t, \omega)}{G(t, \omega)} \tag{11}$$

$G^{h'}(t,\omega)$ denotes the spectrum obtained after deriving the window function. This postprocessing operation results in a higher energy aggregation of the instantaneous frequencies and better frequency resolution.

LMSST is an improved algorithm based on SST that redefines a new frequency operator using the following definition rules:

$$\omega_m(t, \omega) = \begin{cases} \arg\max_{\omega} |G(t, \omega)|, \omega \in [\omega - \Delta, \omega + \Delta], & if |G(t, \omega)| \neq 0 \\ 0, & if |G(t, \omega)| = 0 \end{cases} \tag{12}$$

where, $\Delta$ denotes the discrete frequency interval. When the two signal components reach a frequency of $\varphi'_{k+1}(t) - \varphi'_k(t) > 4\Delta$, at which point the window function reaches a maximum value of zero, the frequency operator is again given a new rule:

$$\omega_m(t, \omega) = \begin{cases} \varphi'_k(t), & if \ \omega \in [\varphi'_k(t) - \Delta, \varphi'_k(t) + \Delta] \\ 0, & otherwise \end{cases} \tag{13}$$

Therefore, LMSST can be expressed as follows

$$LMSST(t, \eta) = \int_{-\infty}^{+\infty} G(t, \omega)\delta(\eta - \omega_m(t, \omega))d\omega \tag{14}$$

## 3. LMSBCT

### 3.1 Theory

Inspired by the LMSST, this study reassigns new time-frequency coefficients in the frequency direction by further processing the SBCT analysis results. According to Eqs (11) and (12), the instantaneous frequency in Eq (11) should be calculated twice through STFT. One is obtained through the conventional STFT, and the other is obtained through STFT, by deriving the window function. To reduce the computational effort, this study used the frequency operator of the local maximum, which only performs the TFA method once.

Therefore, this study proposes a new TFA method, which is expressed as follows:

$$T_{LMSBCT}(t_c, \eta) = \int_{-\infty}^{+\infty} SBCT(f, t_c, a_1, a_2)\delta(\eta - \omega_m(f, t_c))df \tag{15}$$

The signal formula is

$$s(u) = A(u)\exp(j2\pi \int v(u)du) \tag{16}$$

where, $A(u)$ represents the instantaneous amplitude and $\int v(u)du$ represents the instantaneous phase. At this point, the ideal instantaneous frequency is derived from the instantaneous phase as $v(u)$. The Taylor expansion of $v(u)$ can be written as

$$v(u) \approx v(t_c) + v'(t_c)(u - t_c) + \frac{v''(t_c)}{2}(u - t_c)^2 \tag{17}$$

Substituting Eqs (16) and (17) into Eq (9) yields

$$|SBCT(v, t_c, a_1, a_2)| = \left| \int_{-\infty}^{+\infty} \begin{array}{c} A(u)\exp(j2\pi \int v(u)du)\dots \\ h(u - t_c)\exp(-j2\pi\varphi_s(v, t_c, a_1, a_2)) \end{array} du \right|$$

$$= \left| \int_{-\infty}^{+\infty} \begin{array}{c} A(u)h(u - t_c)\exp(j2\pi(u - t_c)^2(\frac{v'(t_c)}{2} - a_1 v(t_c)))\dots \\ \exp(-j2\pi(u - t_c)^3(\frac{v''(t_c)}{6} - a_2 v(t_c))) \end{array} du \right| \tag{18}$$

The coefficients $a_1$ and $a_2$ can be derived based on the following assumptions:

$$\beta_1(i) = -\frac{\pi}{2} + \frac{\pi}{M+1}i, \quad i = 1, 2, 3, \dots, M \tag{19}$$

$$\beta_2(i) = -\frac{\pi}{2} + \frac{\pi}{N+1}i, \quad i = 1, 2, 3, \dots, N \tag{20}$$

M and N must be set in advance, and different sizes means that the chirp rate is divided into different segments from $-\frac{\pi}{2}$ to $\frac{\pi}{2}$. Larger M and N values indicate a higher frequency

resolution. As

$$a_1 \approx \frac{\tan(\beta_1)}{m_0} \tag{21}$$

$$a_2 \approx \frac{\tan(\beta_1)\tan(\beta_2)}{n_0} \tag{22}$$

the presetting of $m_0$ and $n_0$ reduces the computational load. Thus, choosing $\beta_1$ and $\beta_2$ is another problem. According to kurtosis theory, a larger kurtosis indicates a higher energy concentration. Kurtosis can be expressed as follows [23, 30].

$$K = \frac{(\int_0^V SBCT^4(v, t_c, \beta_1, \beta_2)dv)/V}{((\int_0^V SBCT^2(v, t_c, \beta_1, \beta_2)dv)/V)^2} \tag{23}$$

Therefore, the selection rules of optimal $\beta_1$ and $\beta_2$ are as follows.

$$(\beta_1(t_c), \beta_2(t_c)) = \underset{(\beta_1, \beta_2)}{\arg\max}(K) \tag{24}$$

SBCT is defined as

$$
\begin{aligned}
&SBCT(f, t_c, \beta_1, \beta_2) \\
&= \int_{-\infty}^{+\infty} s(u)h(u - t_c)\exp(-j2\pi f \times (u + \frac{\tan\beta_1}{m_0}(u - t_c)^2 + \frac{\tan\beta_1\tan\beta_2}{n_0}(u - t_c)^3))du \quad (25)
\end{aligned}
$$

Therefore, the new frequency operator can be calculated using the following equation:

$$
\omega_m(f, t_c) = \begin{cases} \arg\max_f |SBCT(f, t_c)|, f \in [f - \Delta, f + \Delta], & if|SBCT(f, t_c)| \neq 0 \\ 0, & if|SBCT(f, t_c)| = 0 \end{cases} \tag{26}
$$

## 3.2 Algorithm implementation

```
LMSBCT Algorithm
1. Initialization: Input L; M; N; m₀; n₀.
2. Calculate SBCT:
for i = 1 to M
    for j = 1 to N
    sub-SBCTs(: , : , i, j) ← SBCT(i, j);
    end for
end for
Find (β₁(t_c), β₂(t_c)) = arg max_(β₁,β₂)(K) Output SBTC(f, t_c)
3. Local maximum synchrosequeezing
Calculate ω_m(f, t_c)
Find ω_m(f, t_c) = arg max_f |SBCT(f, t_c)|,
for t = 1 to T
    for f = 1 to F
    η←ω_m(f, t_c)
    T_LMSBCT(t_c, η) ← T_LMSBCT(t_c, η) + SBCT(f, t_c)
    end for
end for
Output T_LMSBCT(t_c, η)
```

## 4. Simulation analysis

In this section, three sets of simulated signals are used to demonstrate the superiority of the proposed TFA method with a good time-frequency aggregation and high time-frequency resolution. The selected comparison algorithms were STFT, SBCT, GLCT, VSLCT, SST, SET, and RM.

### 4.1 Monocomponent signal

To construct a strongly time-varying monocomponent signal, the simulated signal model is considered as follows:

$$S(t) = \sin(2\pi(340t - 2\exp(-2t + 0.4)\sin(14\pi(t - 0.2)))) \tag{27}$$

The sampling frequency was set to 1024 Hz and the sampling time was 1 s. Fig 1 shows the ideal instantaneous frequency of the signal. The frequency of the signal varies with time, and the quantity of instantaneous frequency change decreases as time increases. The signals were processed using various TFA algorithms. Fig 2(A)–2(C) shows the TFDs of the STFT, SBCT, and GLCT algorithms, respectively. The results of these algorithms are energy-dispersive and have a poor frequency resolution, which is not sufficient to satisfy the signal-characteristics analysis. Fig 2(D) and 2(E) shows the processing results of SST and RM, which have higher energy concentrations than the three aforementioned algorithms and can describe the instantaneous frequency of the signal. The algorithm proposed in this paper is a postprocessing procedure for SBCT. The results are shown in Fig 2(F), which effectively improves the concentration of the time-frequency representation and accurately obtains the instantaneous frequency curve.

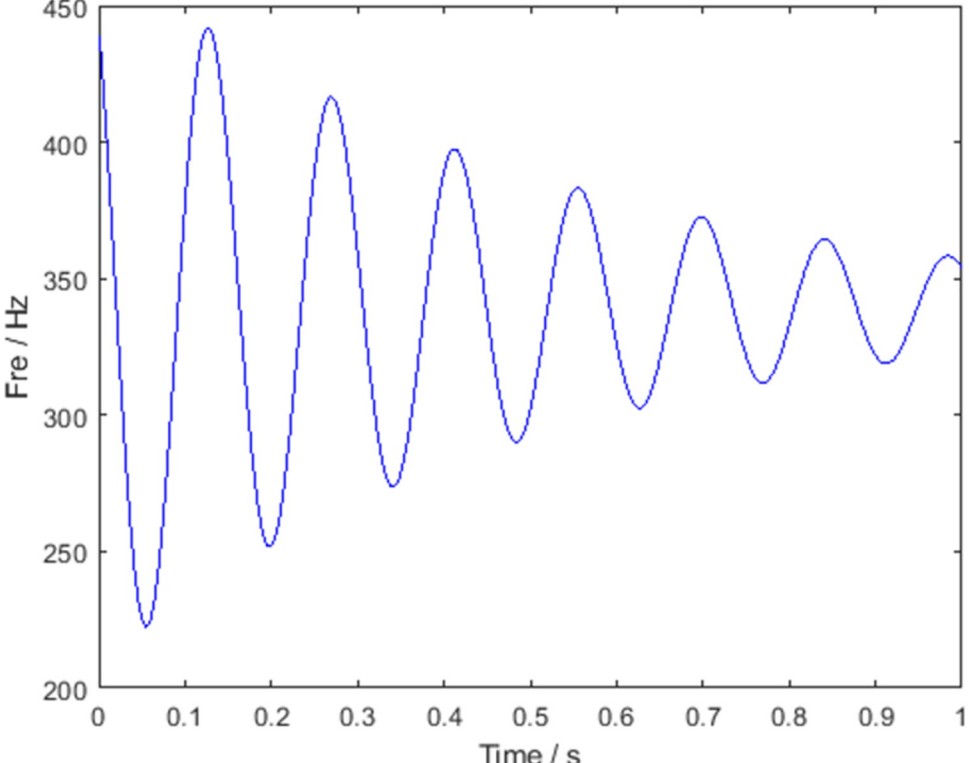

**Fig 1. Ideal instantaneous frequency.**

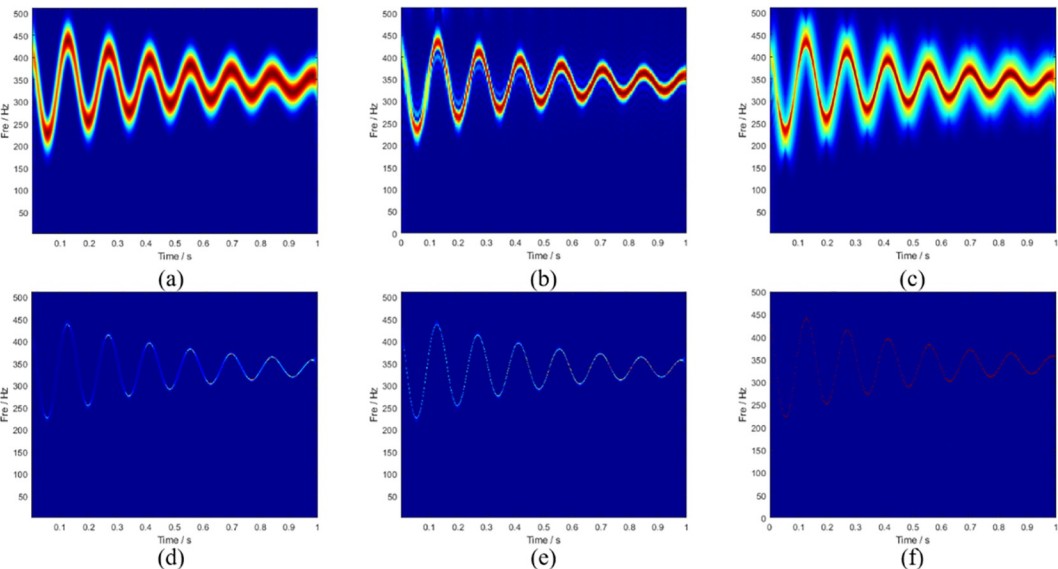

**Fig 2.** TFA results obtained through (a) STFT, (b) SBCT, (c) GLCT, (d) SST, (e) RM, and (f) LMSBCT.

To be able to compare different post-processing results more clearly, Fig 2(D)–2(F) are partially enlarged, and the time between 0.4–0.45 s is selected with ideal frequencies. The red line in Fig 3 represents the ideal instantaneous frequency. The edge of the SST is partially blurred, and the energy aggregation is bad. The result of the RM is slightly blurred. However, a detailed analysis could not be conducted. The improved algorithm proposed in this paper not only has a higher time-frequency resolution but also a better energy focus on the spectrum. It is clearer and more accurate for describing the transient characteristics of the signal.

To objectively evaluate the different TFA methods, this study introduces the concept of Rényi entropy. Entropy is a method of quantitatively evaluating information uncertainty. The more random the signal gets, the greater the uncertainty, and the higher the corresponding entropy value, and vice versa. In the field of TFA, the more concentrated the time-frequency energy distribution, the smaller the uncertainty and the smaller the corresponding entropy value. Therefore, the entropy value can be used to judge the degree of concentration of the time-frequency spectrum energy and evaluate the superiority of the results of different TFA methods [31]. As shown in Table 1, LMSBCT has the smallest Rényi entropy among all methods, which implies the highest time-frequency aggregation. This represents a breakthrough in the field of TFA.

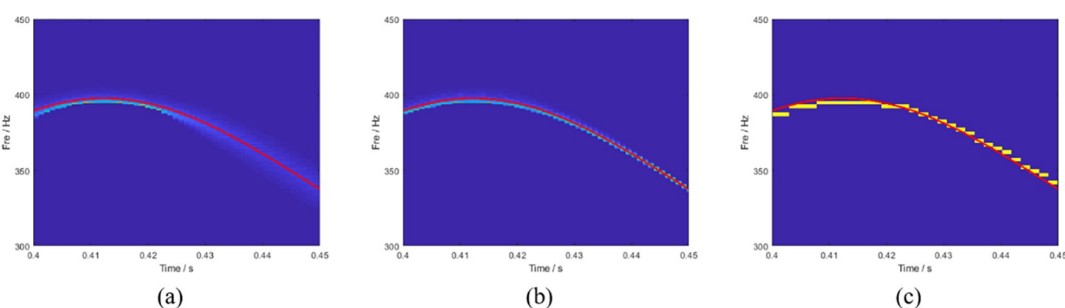

**Fig 3.** (a) Magnified plots of the SST, (b) RM, and (c) LMSBCT results.

**Table 1. Rényi entropy of different algorithms.**

| Method | STFT | SBCT | GLCT | SST | RM | LMSBCT |
|---|---|---|---|---|---|---|
| Rényi entropy | 16.4799 | 12.2216 | 17.0580 | 12.4233 | 11.6416 | 9.6438 |

## 4.2 Multicomponent noise-added signal

To demonstrate the generalizability of the method proposed in this study, the second set of numerically simulated signals consists of multicomponent strong frequency-modulated (FM) signals and noise interference. Under the strong noise interference, the ideal TFA method can effectively identify time-varying features, that is, correctly extract the time-frequency ridges of different components, thus demonstrating its good applicability. The numerical analog signal is specifically represented as

$$S_1(t) = \sin(2\pi \cdot (44 \cdot t + 10 \cdot \sin(t))) \tag{28}$$

$$S_2(t) = \sin(2\pi \cdot (32 \cdot t + 10 \cdot \sin(t))) \tag{29}$$

$$S_3(t) = \sin(2\pi \cdot (10 \cdot t + 2 \cdot \arctan((2 \cdot t - 2)^2))) \tag{30}$$

$$S = S_1(t) + S_2(t) + S_3(t) \tag{31}$$

The ideal instantaneous frequency corresponding to $S_1$, $S_2$, and $S_3$ can be expressed as

$$IF1 = 44 + 10\cos(t) \tag{32}$$

$$IF2 = 32 + 10\cos(t) \tag{33}$$

$$IF3 = 10 + 8 \cdot (2 \cdot t - 2)/(1 + (2 \cdot t - 2)^4) \tag{34}$$

Fig 4 shows the ideal time-frequency ridge and TFD of the multicomponent signal. To test noise robustness, Gaussian white noise was added to the signal, and the calculated signal-to-noise ratio (SNR) was 3 dB. The setting time was 4 s, and the sampling frequency was 120 Hz. Several mainstream TFA methods were compared with the method proposed in this study, and the comparison results are plotted in Fig 5. In terms of the coarseness of the time-

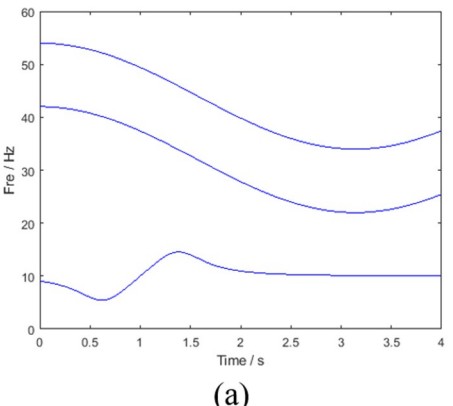
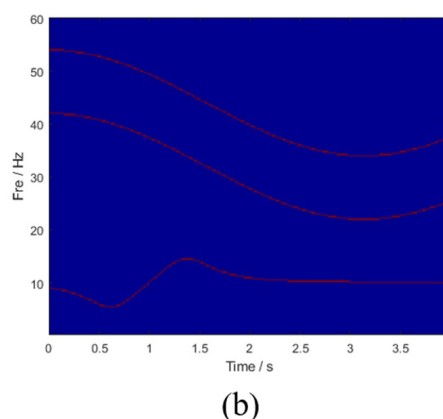

(a)                                                     (b)

**Fig 4.** (a) Ideal instantaneous frequency and (b) ideal TFD.

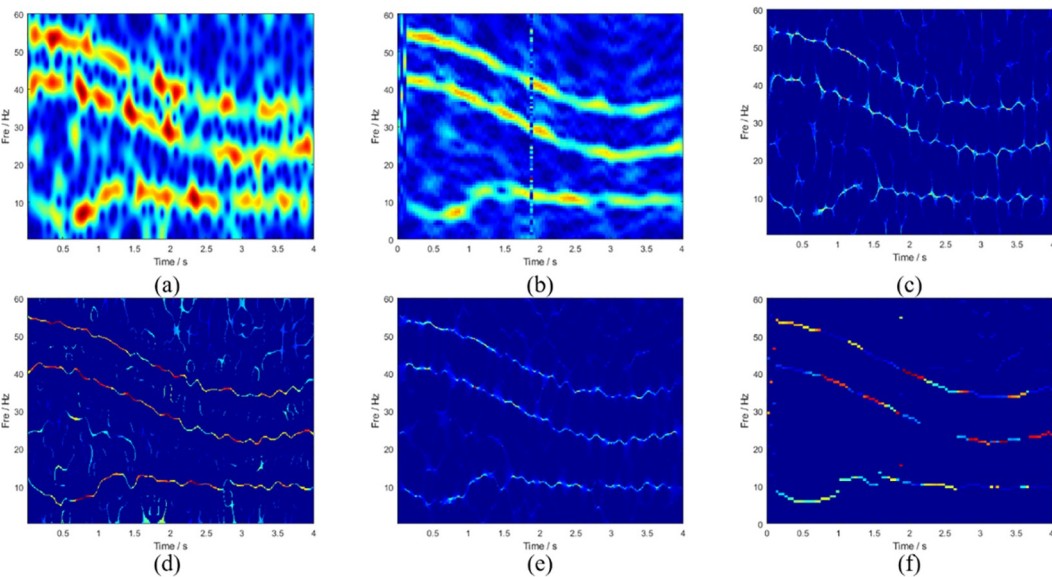

**Fig 5.** TFA results obtained through (a) STFT, (b) SBCT, (c) RM, (d) SET, (e) SST, and (f) LMSBCT.

frequency ridges, the time-frequency energy spread of STFT and SBCT is large, and cannot describe the TFD of the signal well. Although the results of the three methods (SST, SET, and RM processing) are improved, the existence of noise interferes with the identification of the characteristics of the signal. As shown in Fig 5(F), the processing result of LMSBCT has a clean background and no noise interference, which indicates that the method proposed in this study can substantially improve the energy aggregation and has good noise robustness. This is highly consistent with the ideal time-frequency spectrum of the multicomponent simulated signal shown in Fig 4(A) and achieves the optimum time and frequency resolutions. Fig 6 illustrates the instantaneous amplitude spectra of different time-frequency analysis methods at the time of 0.5 s. The spectral bandwidths of STFT and SBCT are large, and there is no boundary between the frequency components. In contrast, the energy of the frequency spectrum of RM and LMSBCT is concentrated in a narrow bandwidth. The frequency component IF3 was locally amplified (represented in red). Comparing Fig 6(C) and 6(D), LMSBCT has clear boundaries between each frequency component, more concentrated energy, and high noise immunity.

To further compare the noise immunity performance of various methods, in this study, the Rényi entropy of different TFA methods was calculated for an input SNR of 0–30 dB. As shown in Fig 7, the Rényi entropy of each method gradually decreases as the SNR gradually increases. The Rényi entropy of SST, SET, and RM are smaller than that of STFT at any SNR, indicating that postprocessing enhances normal time-frequency methods to achieve better resolution. This proves the superiority of the proposed method. In Fig 7, that the Rényi entropy is not significantly affected by the SNR of the input signal, that is, the method in this study has a stronger noise robustness.

## 4.3 Signals with close instantaneous frequency trajectories

The above two simulation experiments prove that the algorithm can obtain the results of TFD with good time-frequency aggregation in strong time-varying signals and multicomponent

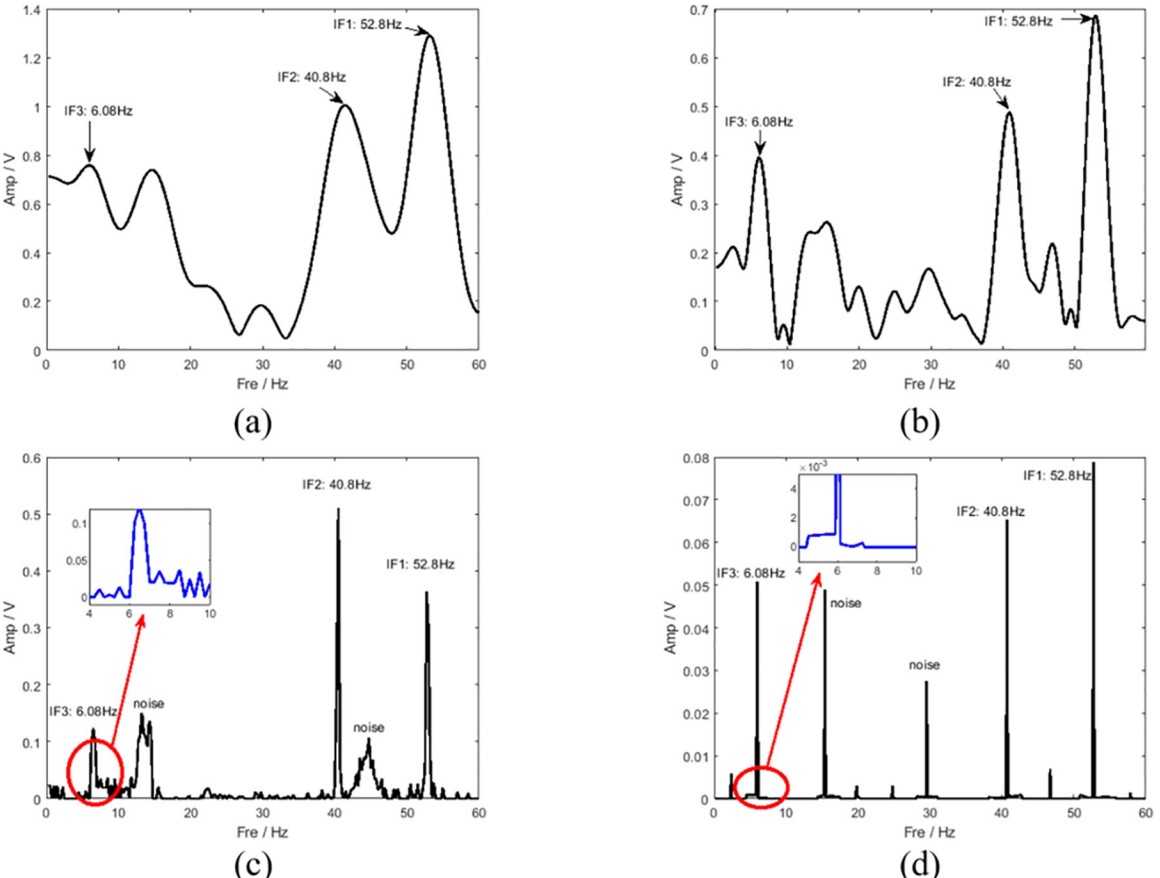

**Fig 6.** TF slices obtained through (a) STFT, (b) SBCT, (c) RM, and (d) LMSBCT at time t = 0.5 s.

signals. The signals in this section are defined by the following equations:

$$S(t) = \sin\left(2\pi \int_0^t v_1(u)du\right) + \sin\left(2\pi \int_0^t v_2(u)du\right) \tag{35}$$

$$v_1(u) = 1/1200 \times (u-45)^2 + 1 \tag{36}$$

$$v_2(u) = 7/7200 \times (u-45)^2 + 7/6 \tag{37}$$

The sampling frequency was 20Hz and the sampling time was 70s. The results of GLCT in Fig 8(A) show a large amount of background noise. The instantaneous frequency ridges of the signal components are completely submerged, and it is impossible to see the number of components present. Fig 8(B) shows the results of STFT, in which the energy dispersion is serious, and the instantaneous frequency trajectory is indistinguishable. Owing to the poor STFT results, the results of the subsequent processing algorithms, SET and SST, also failed to correspond with the expected results. Fig 8(E) and 8(F) presents the analysis results of VSLCT and SBCT, respectively. In the plots, there is no crossover of transient frequencies, but the energy concentration is insufficient. The transient frequency traces in Fig 9 are clear, and there is no cross-mixing. This indicates that even if the signal components are close to each other, the LMSBCT transient frequency division is relatively accurate. Therefore, the energy divergence

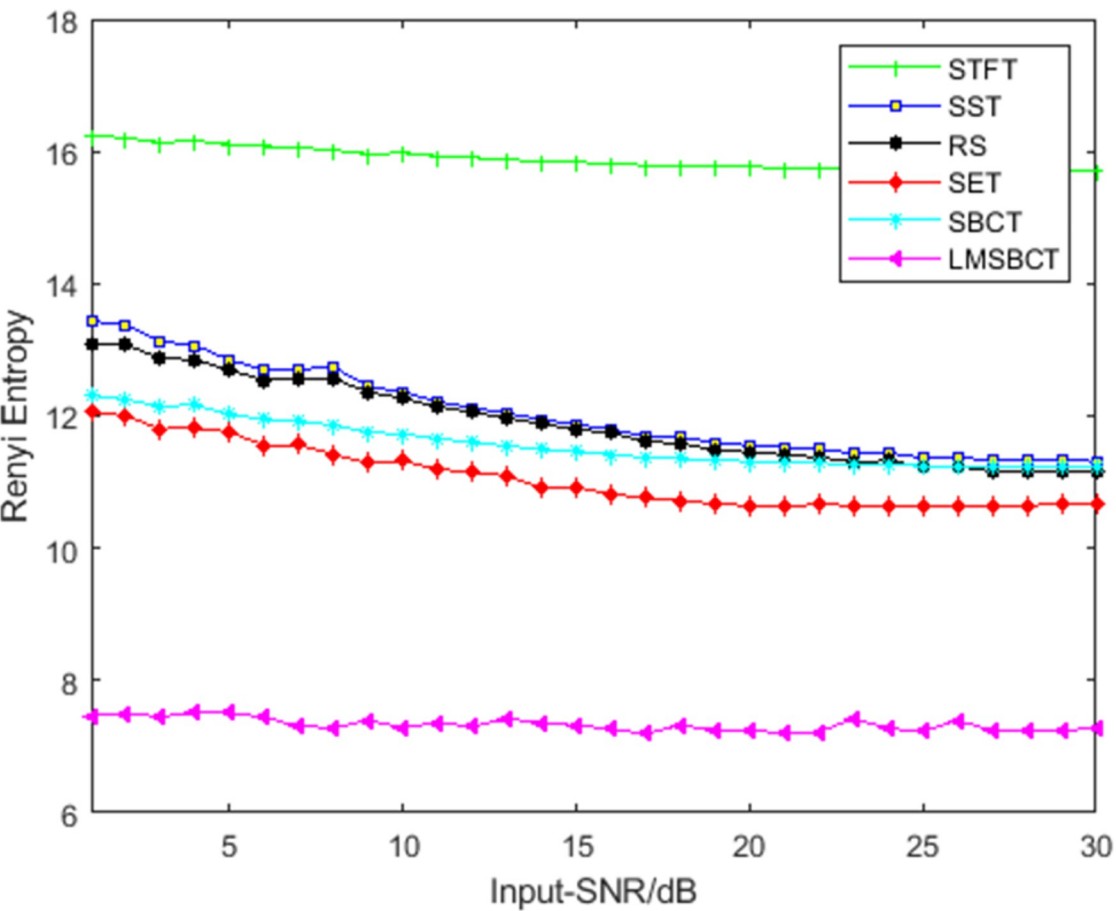

**Fig 7. Plots of the Rényi entropy vs SNR for various TFA methods.**

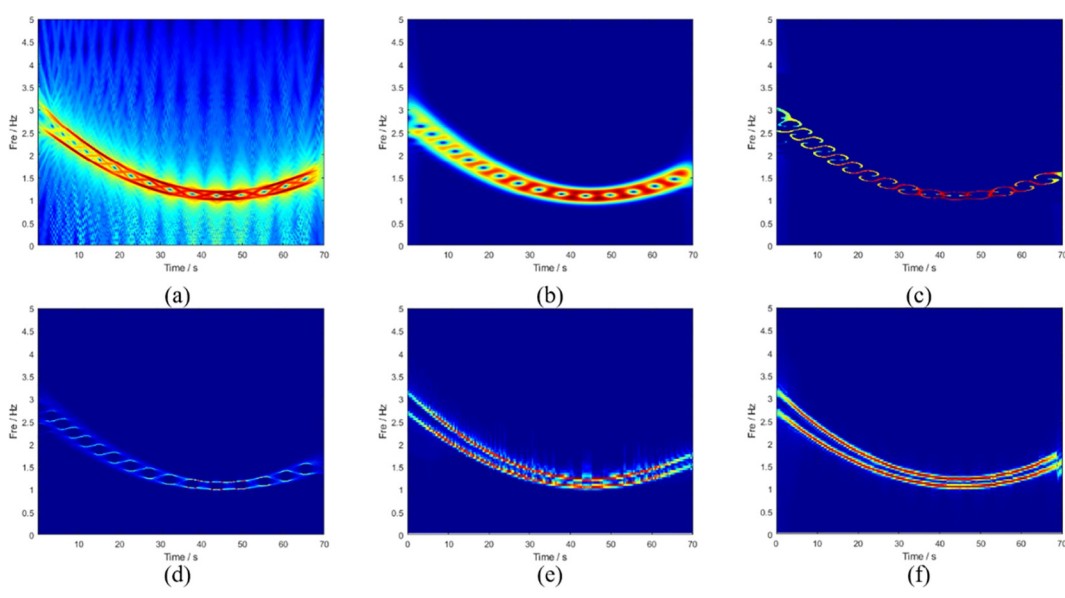

**Fig 8.** TFA results obtained through(a) GLCT,(b) STFT,(c) SET,(d) SST,(e) VSLCT, and (f) SBCT.

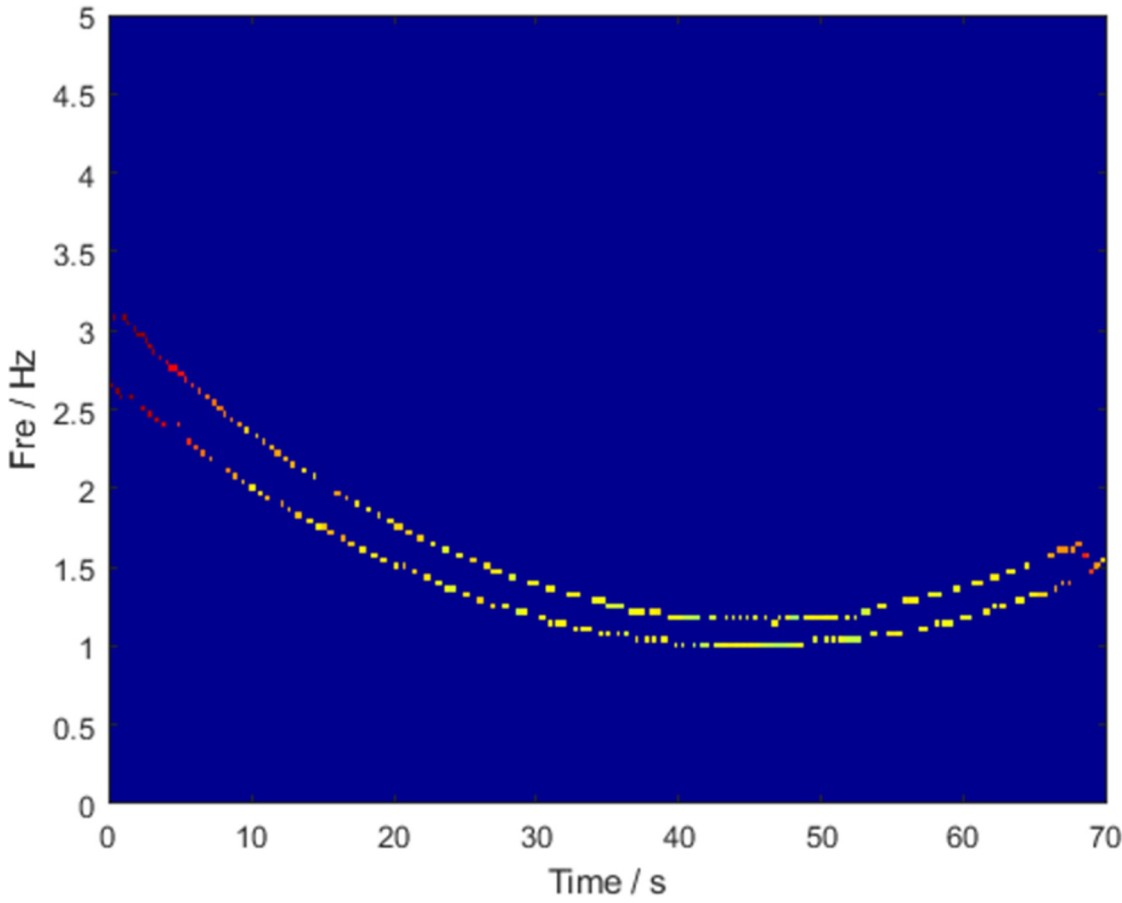

**Fig 9. TFA results obtained through LMSBCT.**

and overlapping phenomena are well-resolved. The improved algorithm proposed in this study has a high time-frequency concentration and is valid for signals with close components, outperforming other typical traditional algorithms. Table 2 shows the Rényi entropy values of the different algorithms, and the proposed algorithm has the smallest Rényi entropy.

## 5. Experimental analysis

### 5.1 Bat signal

The classical bat signal was used as the standard library to validate the method. This signal was first used by Rice University to validate a new method proposed by other researchers [27]. Because the bat signal contains an FM signal, a full down signal, and an echo delay signal, it can effectively verify the results in complex environments. Digitized echolocation pulse emitted by large brown bat *Eptesicus fuscus*. The signal had a sampling point count of 400 and sampling frequency of 140 kHz. It is difficult to accurately understand the nonlinear behavior of bat echolocation in the time-domain signals. Moreover, it is difficult to grasp the time-varying

**Table 2. Rényi entropy of different algorithms.**

| Method | STFT | SBCT | GLCT | SST | SET | VSLCT | LMSBCT |
|---|---|---|---|---|---|---|---|
| Rényi entropy | 15.7705 | 11.4656 | 18.4146 | 13.7068 | 12.9937 | 10.3181 | 8.0927 |

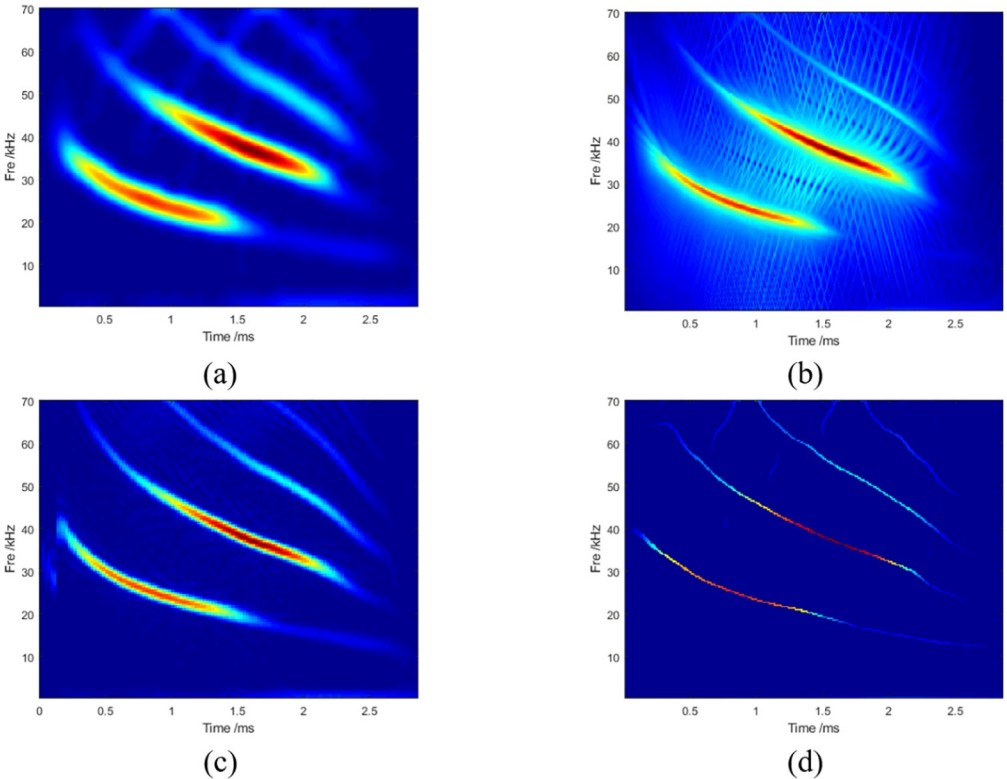

**Fig 10.** TFA results obtained by (a) STFT,(b) GLCT,(c) SBCT, and (d) SET.

characteristics of the signal using only a one-dimensional time-domain or frequency-domain analysis. Extending the one-dimensional time-frequency domain to a two-dimensional time-frequency domain can produce more necessary information. In Fig 10, the time-frequency spectra obtained based on the STFT, GLCT, and SBCT algorithms have a heavy energy divergence, and the resolution is coarse. Noise interferes with the SET method when the frequency is above 60kHz. Fig 11 shows that the LMSBCT substantially improved the effect of the time-frequency spectrum resolution. The energy dispersion phenomenon was resolved effectively. The energy is gathered at the real instantaneous frequency of the signal. In this study, the results of the five TFA methods were also compared using the Rényi entropy as an evaluation index, as shown in Table 3. The Rényi entropy obtained from the LMSBCT method was the smallest, indicating its optimal performance.

## 5.2 Vibration signals from the CWRU dataset

Without loss of generality, the bearing dataset provided by Case Western Reserve University was selected in this study to verify the effectiveness of the proposed algorithm [32]. Experiments were carried out using a 2-horsepower Reliance Electric motor with acceleration data measured near and away from the motor bearings. As shown in Fig 12, the test stand included a 2 hp motor, torque transducer/encoder, dynamometer, and control electronics. Vibration data were collected using accelerometers attached to housing with magnetic bases. It can be observed from Fig 13 that the signal consists of two frequency components. All five TFA algorithms could still roughly identify the instantaneous frequency of the nonstationary signal under the interference of noise, but their noise robustness was different. In Fig 13(A), there is a certain amount of mixing between the two frequency components, which deviates from the

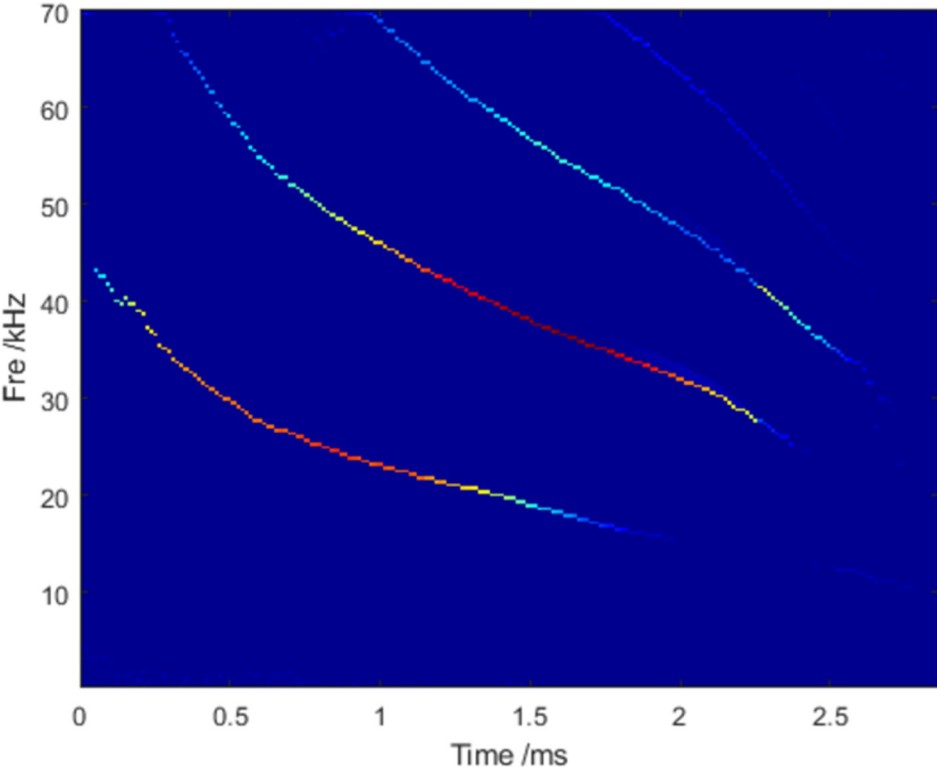

**Fig 11. TFA results obtained by LMSBCT.**

true instantaneous frequency of the multicomponent signal. In the interference of noise, a large number of interwoven noise textures were generated in the background of Fig 13(B) and 13(C). Their identified time-frequency ridges are inlaid in the noise textures. Fig 13(D) shows the approximate instantaneous frequency of the vibration signal at each moment. However, there are some interwoven textures in the background and the originally smooth and continuous time-frequency curve is broken and partially distorted. With the appearance of noise, Fig 13(E) clearly and accurately shows the instantaneous frequency of the signal at each moment. It does not produce excessive noise in the background. Based on the above analysis of the experimental results, the following conclusions can be drawn: Among the five TFA algorithms, the algorithm proposed in this paper is superior.

## 6. Conclusion

Inspired by LMSST, this study proposed a new TFA method, LMSBCT, based on SBCT, which redistributes the new instantaneous frequency operator by extracting the local maxima of the spectrogram in the frequency direction. This method overcomes the shortcomings of traditional TFA methods, improves the aggregation of signals, and achieves high-precision analysis of instantaneous signal frequencies. Three sets of simulation experiments demonstrate the three advantages of this algorithm. (1) The frequency changes of strong time-varying signals

**Table 3. Rényi entropy of different algorithms.**

| Method | STFT | SBCT | GLCT | SET | LMSBCT |
|---|---|---|---|---|---|
| Rényi entropy | 13.9721 | 11.0362 | 15.3418 | 9.8795 | 8.3940 |

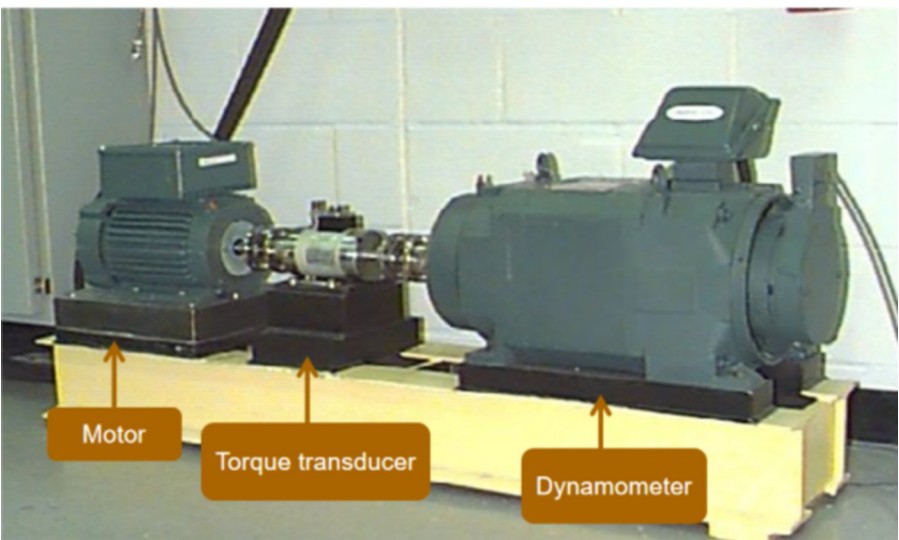

**Fig 12. CWRU teststand.**

can be analyzed effectively. Compared with other TFA methods, the Rényi entropy of LMSBCT can be reduced to 9.6438. (2) The Rényi entropies of the LMSBCT algorithm were always lower than those of the other methods when the SNR was reduced from 30 dB to 1 dB. This implies that the multicomponent signals can be effectively separated, even at low SNRs. (3) This method can also obtain an elaborate TFR when the instantaneous frequencies of the

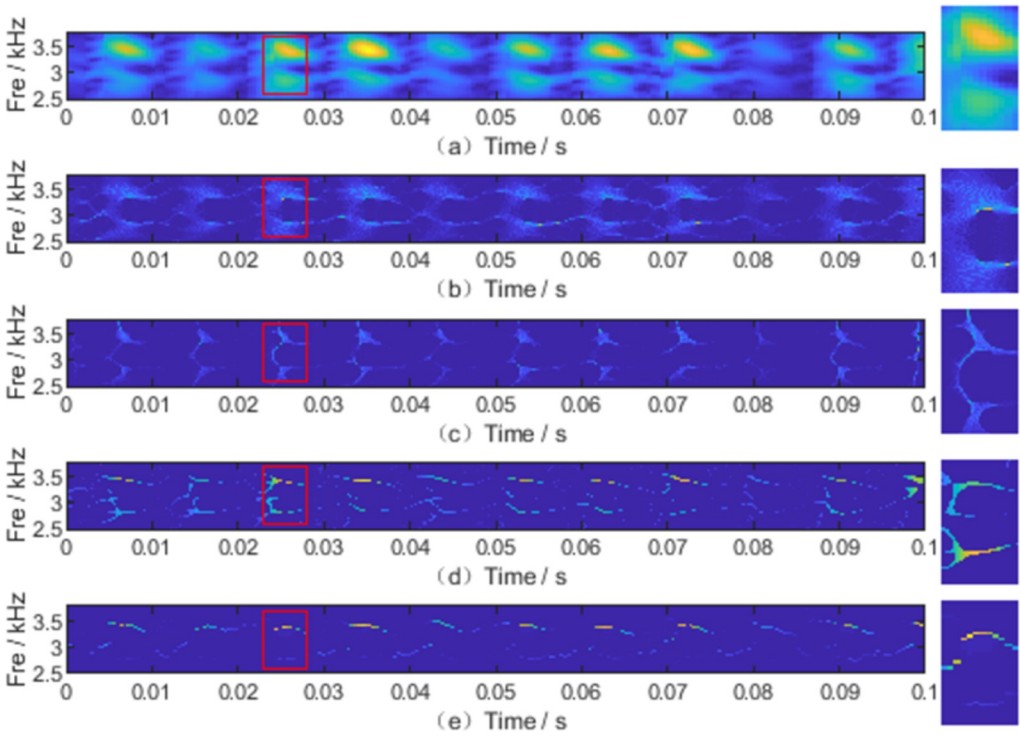

**Fig 13.** TFA results obtained by(a)SBCT, (b)SST, (C)RM, (d)SET, (e) LMSBCT.

signals are close to each other. Even if the frequency interval of the signal is less than 1Hz, the Rényi entropy of the LMSBCT is the smallest compared to the other methods, which is 8.0927. In this study, bat and vibration signals from the CWRU were selected to demonstrate the effectiveness of the algorithm. Compared with other methods, the algorithm in this study can describe and characterize the time-varying features accurately and in detail, which is beneficial for the subsequent extraction and analysis of signal features. Both the simulation signal and the actual vibration signal application results show that the method has the advantages of high time-frequency resolution and good energy aggregation.

## Author Contributions

**Investigation:** Xingcheng Han.

**Methodology:** Yating Hou.

**Validation:** Liming Wang.

**Visualization:** Xiuli Luo.

**Writing – original draft:** Yating Hou.

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
