## [Decision Letter · Decision Letter 0]

25 Oct 2022

PONE-D-22-22608Local maximum synchrosqueezes form scaling-basis chirplet transformPLOS ONE

Dear Dr. hou,

Thank you for submitting your manuscript to PLOS ONE. After careful consideration, we feel that it has merit but does not fully meet PLOS ONE’s publication criteria as it currently stands. Therefore, we invite you to submit a revised version of the manuscript that addresses the points raised during the review process.

We look forward to receiving your revised manuscript.

Kind regards,

Bashar Ibrahim

Academic Editor

PLOS ONE

Journal Requirements:

“The manuscript is funded by:(1)Science  and Technology Innovation Project of Colleges and Universities in Shanxi Province(China),the award number is 2020L0301;（2）Fundamental Research Program of Shanxi Province(China), the award number is 20210302124545.”

5. Please ensure that you refer to Figure 10 in your text as, if accepted, production will need this reference to link the reader to the figure.

Reviewers' comments:

Reviewer's Responses to Questions

**Comments to the Author**

1. Is the manuscript technically sound, and do the data support the conclusions?

Reviewer #1: Yes

Reviewer #2: Yes

2. Has the statistical analysis been performed appropriately and rigorously? 

Reviewer #1: Yes

Reviewer #2: Yes

3. Have the authors made all data underlying the findings in their manuscript fully available?

Reviewer #1: Yes

Reviewer #2: Yes

4. Is the manuscript presented in an intelligible fashion and written in standard English?

Reviewer #1: Yes

Reviewer #2: Yes

5. Review Comments to the Author

Reviewer #1: The paper proposes a new TFA method to obtain better time-frequency representation for nonstationary signals. This topic is important in signal processing and its real-life applications, and the presented transform is interesting. Theoretical basis is well described with detailed proof. The data validation shows that the proposed method has excellent performance for the signal with strongly time-varying IF, by comparing to several classical time-frequency methods.

1. In order to better compare the TF concentration of the TFR obtained different TF methods, it is suggested to display the RE values of each component in the table as EMD.

2. The authors should introduce more details of the experimental device and the experimental implementation, rather than just give the data source.

7.In Section 6, it is suggested that the authors also use quantitative indicators to measure the TFR results of different TF analysis methods and analyze them.

Reviewer #2: 1 The authors shoud state the contribution and the novelty of this paper. The proposed method LMSBCT is a simple combination of SBCT and LMSST.

2 The English level of this paper should be improved.

6. PLOS authors have the option to publish the peer review history of their article (what does this mean?). If published, this will include your full peer review and any attached files.

Reviewer #1: No

Reviewer #2: No

---

## [Author Response · Author response to Decision Letter 0]

7 Nov 2022

We would like to thank the reviewers and the editor for their time in reviewing our paper and suggestions for improving it. We have written a detailed point-by-point response to the reviewers to address their comments and explain the corresponding changes made in the manuscript.

The detailed point-by-point responses to journal requirements are given as follows:

(1) We amend our manuscript to meet PLOS ONE's style requirements according to the PLOS ONE style templates;

(2) In financial disclosure, the funders assisted with study design and manuscript preparation;

(3) We amend our Data Availability Statement and provide the corresponding URLs where the datasets can be accessed;

The Bat signal used in this paper came from Rice University. The website is https://web.archive.org/web/20160403234536/http://dsp.rice.edu/software/bat-echolocation-chirp.The Vibration signals from Case Western Reserve Univesity dataset. The website is https://engineering.case.edu/bearingdatacenter/apparatus-and-procedures.

(4) We have created an ORCID iD and updated my Information;

(5) We were sorry for our careless mistakes. We ensure that all figures were mentioned in the manuscript.

We would like to thank the reviewers for their valuable comments, which have helped us to further improve the contribution of the paper. We have addressed all the comments and suggestions and revised our manuscript as follows:

For Reviewer: 1

Response to the comments

Comment 1: In order to better compare the TF concentration of the TFR obtained different TF methods, it is suggested to display the RE values of each component in the table as EMD. 

Our Response: The authors sincerely appreciate the valuable comments. EMD is to divide the nonstationary signal into several intrinsic mode functions from high to low frequencies, and then perform Hilbert transform on each IMF component of the decomposition. The method used in this study is to choose a short window for the signal to be observed. It is assumed that the signal within each short window is stationary, but each window may contain more than one prominent frequency component. It is difficult to evaluate Rényi entropy for each component. In order to better evaluate the processing performance of different methods for each frequency component, this paper shows the instantaneous amplitude spectra of several methods at the time of 0.5s.

In Section 4.2, figure 6 was added [p17, line293].In the third paragraph of Section 4.2, the following sentences have been added to explain Fig 6 and improve the quality of the paper [p16, line285-292]:

Fig 6 illustrates the instantaneous amplitude spectra of different time-frequency analysis methods at the time of 0.5 s. The spectral bandwidths of STFT and SBCT are large, and there is no boundary between the frequency components. In contrast, the energy of the frequency spectrum of RM and LMSBCT is concentrated in a narrow bandwidth. The frequency component IF3 was locally amplified (represented in red). Comparing Figs 6(c) and (d), LMSBCT has clear boundaries between each frequency component, more concentrated energy, and high noise immunity.

Comment 2: The authors should introduce more details of the experimental device and the experimental implementation, rather than just give the data source.

Our Response: The authors sincerely appreciate the valuable comments. The authors have checked the literature carefully and added more details about the experimental device and the experimental implementation in section 5 of the revised manuscript.

Actions on Manuscript: 

In Section 5.1, “Digitized echolocation pulse emitted by the Large Brown Bat, Eptesicus Fuscus.” was added [p20, line339-340]. 

To better show the effectiveness of the algorithm in this paper, the authors changed this part of the processed signal to a multicomponent bat echo signal. The detailed modification can be found in 'Response to Reviewers'.

In Section 5.2, The authors added the experimental setup diagram [p24, line383-384] and the data acquisition method [p23, line362-367]. The following sentences have been added to the manuscript: “Experiments were carried out using a 2-horsepower Reliance Electric motor with acceleration data measured near and away from the motor bearings. As shown in Fig 12, the test stand included a 2 hp motor, torque transducer/encoder, dynamometer, and control electronics. Vibration data were collected using accelerometers attached to housing with magnetic bases”.

Comment 3: In Section 6, it is suggested that the authors also use quantitative indicators to measure the TFR results of different TF analysis methods and analyze them.

Our Response: The authors sincerely appreciate the valuable comments. The authors have rewritten this part according to the Reviewer’s suggestion. 

Actions on Manuscript: The authors again use the Rényi entropy indicators to highlight the superiority of the proposed algorithm. This action will make the conclusions more objective and credible. The revised paragraph[p25, line396-403] reads as follows：

(1) The frequency changes of strong time-varying signals can be analyzed effectively. Compared with other TFA methods, the Rényi entropy of LMSBCT can be reduced to 9.6438. (2) The Rényi entropies of the LMSBCT algorithm were always lower than those of the other methods when the SNR was reduced from 30 dB to 1 dB. This implies that the multicomponent signals can be effectively separated, even at low SNRs. (3) This method can also obtain an elaborate TFR when the instantaneous frequencies of the signals are close to each other. Even if the frequency interval of the signal is less than 1Hz, the Rényi entropy of the LMSBCT is the smallest compared to the other methods, which is 8.0927.

For Reviewer: 2

Response to the comments

Comment 1: The authors should state the contribution and the novelty of this paper. The proposed method LMSBCT is a simple combination of SBCT and LMSST.

Our Response: The authors sincerely appreciate the valuable comments. LMSST is essentially a post-processing method of STFT. Fundamentally, the effectiveness of these post-processing methods and the superiority of the processed signals also depend on the original time-frequency analysis method. SBCT is also an improved method of STFT. Our manuscript proposes the LMSBCT method by making a good combination of SBCT and LMSST for the first time based on the improvement of STFT, i.e., SBCT. Such an approach would give the results of time-frequency analysis the advantages of SBCT as well as the advantages of increased post-processing. Simulated and real-life signals are employed to validate the advantages of LMSBCT by comparing them with some advanced TF methods. The experimental results demonstrate that LMSBCT can provide a better time-varying description, obtain a more precise IF estimate, and have stronger noise robustness.

Actions on Manuscript: The authors have added a sentence to the Abstract section to clarify the novelty of this paper. The added sentence is shown below [p1, line17-18]:

which is a further improvement of the scaling-basis chirplet transform (SBCT) with energy rearrangement in frequency and can be viewed as a good combination of SBCT and local maximum synchrosqueezing transform. 

Comment 2: The English level of this paper should be improved.

Our Response: The authors sincerely appreciate the valuable comments. The authors revised and improved the English of our manuscript. The authors have polished our article by editage.

Thanks again for your thoughtful suggestion, it is very important. Due to your suggestion, we found some shortcomings in our current work. We will improve our research level according to your suggestion in future work and get more achievements. We are hoping to learn more from you.

---

## [Decision Letter · Decision Letter 1]

14 Nov 2022

Local maximum synchrosqueezes form scaling-basis chirplet transform

PONE-D-22-22608R1

Dear Dr. hou,

We’re pleased to inform you that your manuscript has been judged scientifically suitable for publication and will be formally accepted for publication once it meets all outstanding technical requirements.

Kind regards,

Bashar Ibrahim

Academic Editor

PLOS ONE

Additional Editor Comments (optional):

Reviewers' comments:

Reviewer's Responses to Questions

**Comments to the Author**

1. If the authors have adequately addressed your comments raised in a previous round of review and you feel that this manuscript is now acceptable for publication, you may indicate that here to bypass the “Comments to the Author” section, enter your conflict of interest statement in the “Confidential to Editor” section, and submit your "Accept" recommendation.

Reviewer #2: All comments have been addressed

2. Is the manuscript technically sound, and do the data support the conclusions?

Reviewer #2: Yes

3. Has the statistical analysis been performed appropriately and rigorously? 

Reviewer #2: Yes

4. Have the authors made all data underlying the findings in their manuscript fully available?

Reviewer #2: No

5. Is the manuscript presented in an intelligible fashion and written in standard English?

Reviewer #2: Yes

6. Review Comments to the Author

Reviewer #2: No further comment. The authors have addressed all my concerns. This paper is good and can be accepted.

7. PLOS authors have the option to publish the peer review history of their article (what does this mean?). If published, this will include your full peer review and any attached files.

Reviewer #2: No

---

## [Editor Report · Acceptance letter]

17 Nov 2022

PONE-D-22-22608R1 

Local maximum synchrosqueezes form scaling-basis chirplet transform 

Dear Dr. Hou:

I'm pleased to inform you that your manuscript has been deemed suitable for publication in PLOS ONE. Congratulations! Your manuscript is now with our production department. 

Kind regards, 

on behalf of

Prof. Dr. Bashar Ibrahim 

Academic Editor

PLOS ONE